# Current Evidence on the Use of Hyaluronic Acid as Nonsurgical Option for the Treatment of Peyronie's Disease: A Contemporary Review

Marco Capece [1] , Giuseppe Celentano [1],* and Roberto La Rocca [2]

1  Unit of Urology, Department of Neurosciences, Reproductive Sciences and Odontostomatology, University of Naples "Federico II", 80131 Naples, Italy
2  Department of Neurosciences, Reproductive Sciences and Odontostomatology, University of Naples "Federico II", 80131 Naples, Italy
*  Correspondence: dr.giuseppecelentano@gmail.com; Tel.: +39-338-968-27

**Abstract:** Peyronie's disease is a condition characterized by the formation of fibrous plaques in the tunica albuginea, which can cause pain, curvature, and erectile dysfunction. Preclinical studies have demonstrated the potential benefits of hyaluronic acid in treating Peyronie's disease, including antifibrotic, anti-inflammatory, and proangiogenic effects, although more research is needed to fully understand its mechanisms of action. Clinical studies have shown promising results, with hyaluronic acid injections leading to improvements in plaque size, penile curvature, and erectile function, and being well tolerated by patients. The findings suggest that HA injections could be a viable and safe treatment option for Peyronie's disease, particularly in the early stages of the disease. However, more research is needed to determine the optimal dosage and treatment duration for HA injections, and to confirm its efficacy in the stable phase of Peyronie's disease. Overall, hyaluronic acid is a potentially effective therapy for Peyronie's disease, with the ability to inhibit fibrosis and promote angiogenesis, and low risk of adverse effects, making it an attractive option for patients who are unable or unwilling to undergo surgery.

**Keywords:** hyaluronic acid; Peyronie's disease; intraplaque injections; induration penis plastica; penile curvature

## 1. Introduction

Peyronie's disease (PD) is a connective tissue disorder that affects up to 10% of men worldwide, with a peak incidence between the ages of 45 and 60 years [1]. PD is characterized by fibrous plaque formation in the tunica albuginea of the penis, which leads to penile curvature, deformity, and penile pain during erections. The exact etiology of PD is still unclear, although it is thought to involve a combination of genetic, vascular, and mechanical factors [2]. Regardless of the etiology involved, there is currently no definitive cure for PD, and the available treatment options are limited and sometimes associated with adverse effects. Surgery is usually considered as a last resort for patients with severe PD and significant functional impairment [3].

Nonsurgical treatments for Peyronie's disease include a variety of options, such as oral antioxidant therapies or intralesional injections. Antioxidants, such as vitamin E and coenzyme Q10, have been suggested to potentially reduce oxidative stress and inflammation, which are believed to play a role in the progression of Peyronie's disease. However, there is limited research specifically addressing the use of antioxidants in Peyronie's disease [4]. Regarding intralesional injections, collagenase clostridium histolyticum (CCH) is an example of a nonsurgical option that has gained attention in recent years [5]. CCH is an enzyme that breaks down collagen, the protein responsible for the fibrous plaques that develop in Peyronie's disease. It is injected directly into the plaque and has been shown

to improve penile curvature and reduce symptoms in clinical trials [5,6]. However, CCH treatment can also be associated with adverse effects such as swelling, bruising, and penile hematoma [6].

In recent years, hyaluronic acid (HA) has emerged as a novel therapy for PD due to its ability to modulate inflammation, promote tissue repair, and restore extracellular matrix homeostasis [7]. HA is a natural glycosaminoglycan that is found in various tissues of the body, including the penis. It plays an important role in maintaining the structural integrity and function of the extracellular matrix by regulating cell proliferation, differentiation, and migration [7]. HA has also been shown to have anti-inflammatory, antioxidant, and analgesic effects, which may be beneficial in the management of PD [8].

Despite the potential benefits of HA in PD, the current evidence on its efficacy and safety is still limited and inconclusive. Several preclinical and clinical studies have investigated the use of HA in PD, creating new perspectives for the nonsurgical treatment of PD [9]. For example, a recent review by Schifano et al. (2021) reported that although some studies have shown promising results with HA for PD, the overall quality of the evidence is low and more high-quality studies are needed to establish its efficacy [10].

In this review, we aim to critically evaluate the current evidence on the use of HA in the management of PD. By synthesizing the available data, we hope to provide valuable insights into the potential use of HA as a therapeutic option for PD, evaluate other nonsurgical therapies for different phases of PD, and identify the key research gaps and future directions in this area.

## 2. Materials and Methods

This narrative review was conducted by searching electronic databases (PubMed, Embase, and Cochrane Library) using the following keywords: "Peyronie's disease", "hyaluronic acid", "intralesional injection", "erectile function", "penile curvature", and "sexual satisfaction". Only studies published in English from 1990 to 2023 were included. The search yielded a total of 35 studies, and their titles and abstracts were screened to exclude irrelevant studies. The full text of the remaining studies was then reviewed to identify relevant studies that met the inclusion criteria. In addition, reference lists of relevant studies and review articles were searched for additional studies that were not identified through the electronic search. Data from the included studies were extracted and synthesized in a narrative format.

## 3. Mechanisms of Action of Hyaluronic Acid in Peyronie's Disease

HA is a naturally occurring glycosaminoglycan that has been found to play a role in tissue repair and regeneration. In PD, HA has been proposed as a potential treatment option due to its ability to modulate the inflammatory response and promote tissue healing. This section will review the proposed mechanisms of action of HA in PD, including its effects on fibrosis, inflammation, and angiogenesis.

Several studies have suggested that HA can inhibit fibrosis by suppressing the production of collagen and other extracellular matrix components [11]. Fibrosis is a pathological process characterized by the excessive deposition of extracellular matrix (ECM) proteins, including collagen, in response to tissue injury or chronic inflammation. It can lead to the formation of fibrotic plaques in the tunica albuginea of the penis, resulting in penile curvature and erectile dysfunction in PD patients.

The molecular pathways underlying the inhibitory effects of HA on fibrosis are not fully understood, but some studies have proposed several mechanisms. HA has been shown to interact with cell surface receptors, such as CD44 and RHAMM, to modulate cellular functions and signaling pathways involved in fibrosis [12]. Moreover, HA can inhibit the TGF-β/Smad signaling pathway, which is a major pathway involved in the production of ECM proteins, by downregulating the expression of TGF-β receptors and Smad proteins [13].

Regarding the anti-inflammatory effect, HA is involved in two basic mechanisms that determine its biological functions. Firstly, it acts as a structural molecule by modulating the tissue hydration, osmotic balance, and physical properties of ECM, where it creates a hydrated and stable space for the maintenance of cells, collagen and elastin fibers, and other ECM components. Secondly, HA acts as a signaling molecule when interacting with its binding molecules. The effects of HA are dependent on its molecular weight, location, and specific cell factors such as receptor expression, signaling pathways, and cell cycle. HA and its associated proteins can promote or inhibit inflammation, cell migration, activation, division, and differentiation, depending on these factors [14].

In addition to its effects on fibrosis and inflammation, HA has been shown to promote angiogenesis. The mechanism is not fully understood, but several studies have suggested that it may involve the upregulation of growth factors and cytokines that stimulate angiogenesis. One possible mechanism is the activation of vascular endothelial growth factor (VEGF), a potent angiogenic factor that promotes the growth of new blood vessels [15].

## 4. Clinical Studies on the Use of Hyaluronic Acid in Peyronie's Disease

Clinical studies on the use of HA in PD have shown promising results. Five studies have been designed to evaluate the effect of HA in PD.

Gennaro et al. assessed the efficacy of injectable HA as a local therapy for the acute phase of PD. A total of 83 PD patients received 30 penile infiltrations with 20 mg HA over the course of 6 months, while 81 PD patients were left without any therapy. Follow-up examinations were undertaken after the conclusion of therapy and 12 and 24 months later. All treated PD patients exhibited a reduction in plaque size, an improvement in penile curvature, and an improvement in penile stiffness, with an average rise of 21.1% in the IIEF score at the 12-month follow-up. The stability of these benefits was maintained during the 24-month follow-up. The authors found that intralesional injections of HA into the penile tissue are an effective treatment for PD [16].

Therefore, Zucchi et al. designed a prospective, single-arm, self-controlled, interventional, multicenter pilot study to evaluate the efficacy of intralesional injections of HA in patients with early phase of PD. Sixty-five patients received a ten-week cycle of weekly intraplaque injections of HA and were reassessed two months following the conclusion of treatment. The primary outcome measures were plaque size, penile curvature, the IIIEF-5 score, the VAS score for sexual pleasure, and the Patient's Global Impressions of Improvement (PGI-I) score. Post-treatment improvements in plaque size, penile curvature, IIEF-5 score, and VAS score were statistically significant. Total PGI-I questionnaire improvement was 69%. Intralesional therapy with HA can reduce plaque size, penile curvature, and overall sexual pleasure, and appears to be most appropriate in the early (active) phase of illness [17].

In 2017, another study aimed to compare the efficacy of intraplaque injection of verapamil (ILVI) and HA in treating early onset PD. Sexually active men aged 18 and above were randomly assigned to receive either ILVI or HA injections weekly for 12 weeks. The primary outcome measured was the change in penile curvature, with secondary outcomes including changes in plaque size and IIEF-5 score. The study found that there was no significant difference in plaque size or IIEF-5 score between the two groups. However, there was a significant decrease in penile curvature in the HA group compared to the ILVI group. Additionally, patients in the HA group reported a greater improvement in the PGI-I score. Overall, the study suggested that intralesional HA may be more effective than ILVI in treating PD in terms of penile curvature and patient satisfaction [18].

Cocci et al. compared the effectiveness and safety of intralesional injections of HA versus verapamil in patients with PD in the acute phase. A total of 244 patients were included in the study, with 125 receiving HA injections and 119 receiving verapamil injections. After 8 weeks of treatment, penile curvature decreased more significantly in the HA group than the verapamil group. Additionally, the HA group had greater improvements in the IIEF-15 score and VAS than the verapamil group. The study suggests

that intralesional HA injections could be an effective and safe treatment option for patients with acute-phase PD [19].

Another Italian group presented the first prospective, randomized phase III clinical study comparing the efficacy of a combination of oral administration and intralesional injection of HA to intralesional injections alone in individuals with an active phase of PD. Two groups of patients were randomly assigned. Group A received the oral administration of HA in addition to weekly intralesional injections of HA for 6 weeks, while Group B only received weekly intralesional injections for 6 weeks. In comparison to Group B, Group A saw a much greater decrease in penile curvature and a greater improvement in IIEF-5 and PGI-I scores. The research finds that oral administration in conjunction with intralesional HA therapy is more effective at enhancing penile curvature and overall sexual pleasure than intralesional HA treatment alone [20].

## 5. Safety and Adverse Effects of Hyaluronic Acid for Peyronie's Disease

As with any medical treatment, it is important to consider the safety and potential adverse effects of HA for the treatment of PD. Fortunately, HA has a well-established safety profile with minimal side effects. In fact, intralesional injection of HA may be considered at minimal risk of adverse events (AE).

One of the most common adverse effects reported with HA injections is minor bruising or redness at the injection site, but these effects are also typically mild and resolve quickly [16]. These AE have been reported in only one study, whereas in the others, no significant adverse effects were reported nor have injection-site ecchymosis/hematomas been recorded [17–20].

There have been no reports of serious adverse events associated with the use of HA for PD. However, as with any medical intervention, there is a potential risk of infection. In order to minimize this risk, it is important to follow proper sterile techniques during the injection procedure [16–20].

## 6. Nonsurgical Alternatives to Hyaluronic Acid for Peyronie's Disease

To date, many nonsurgical options are present and may be used. Interferon-2b intraplaque injections are one of the possibilities. Interferon-2b (IFN—2b) has been shown to decrease fibroblast proliferation and formation of collagen and other extracellular matrix (ECM) proteins by boosting collagenase levels and decreasing metalloproteinases, which inhibit collagenase [21]. These properties of IFN-$\alpha$-2b have led to its widespread use in the treatment of hypertrophic scars, liver fibrosis, and other fibrotic conditions resulting from fibroblast dysregulation [22]. In a study by Stewart et al. [23], intralesional treatment of interferon-2b resulted in a greater than 20% decrease in penile curvature and a total response rate of 91%, regardless of the location of PD plaque. Likewise, Trost et al. [24] observed equivalent results after intralesional injections of IFN-2b in patients suffering from a curvature of less than 30 degrees, without noticing any changes in penile vascular parameters. Sokhal et al. reported in a prospective trial a substantial improvement in plaque volume and penile curvature after intralesional IFN-2b therapy [25].

Another enzyme that has been tested and proved effective in the last decade is collagenase clostridium hystoliticum (CCH). CCH is a combination of class-I and class-II clostridial collagenases (AUX-I and AUX-II) that exhibit similar and complementary substrate specificity, making it effective in breaking down the fibrotic composition of PD plaques and collagen types I and III [26]. Two large RCTs in phase III, Investigation for Maximal Peyronie's Reduction Efficacy and Safety Studies (IMPRESS) I and II, conducted in 2010, demonstrated a significant improvement in curvature deformity and PD bother domain score of the PD questionnaire (PDQ) [27]. Consequently, in 2013, the Food and Drug Administration (FDA) approved the intralesional injection of CCH in order to treat adult PD patients with PD, palpable plaque, and penile curvature $\geq 30°$ [28]. Since then, many studies have been performed evaluating the efficacy of CCH injections and many

protocols have been standardized. No studies have been designed in order to compare such protocols [5,29,30].

Verapamil has been used for a number of decades after its effectiveness in animals was shown. It has been proven that intralesional injection of verapamil into a rat model of Parkinson's disease results in a reduction in plaque size, penile curvature, and levels of collagen and elastin [31]. In the oldest human trial, using intraplaque injection of verapamil, which was later evaluated by other researchers, 14 participants were given injections every 2 weeks for a period of 6 months, with the dosage eventually reaching 10 milligrams per each injection. The plaque was reported to have become softer by all of the individuals, and the penile constriction and curvature were both reported to have improved by 43 percent [32]. In a follow-up study, 38 men who had completed the full treatment course of 10 mg intralesional injections every other week for a total of 12 injections showed that pain had been eliminated in almost all cases. Additionally, 76% of the men had a subjective improvement in curvature, and 72% reported improvement in their ability to engage in penetrative intercourse. A total of 54% of males saw a reduction in the amount of curvature in their spines [33]. These encouraging findings led researchers to conduct a bigger trial with a total of 140 male participants, each of whom received an intralesional injection of 10 mg of verapamil. According to the findings of the study, 62% of males had a reduction in their curvature of 17 degrees or more. A total of 83% of males had an improvement in their waist circumference, 80% saw an improvement in their stiffness distal to the plaque, and 71% saw an improvement in their sexual function [34]. On the other hand, the effects of numerous additional investigations with intralesional verapamil have not been proven to be as strong [35,36]. It is important to note that Cavallini and colleagues later found that the optimal concentration for verapamil was 10 mg in 20 mL of injectable solution, which maximized penile curvature improvement, the size of the plaque, IIEF, and pain [36].

### 7. Potential Novel Intralesional Treatments

At this point in time, the focus is on mesenchymal stem cells (MSC), which are being closely examined. Because of the potential of MSC in reducing fibrosis, there has been an uptick in interest in investigating their use in the treatment of PD. In rat models of tunica albuginea fibrosis, a subtype of mesenchymal stem cells known as adipose-derived stem cells (ADSC) has been studied. The first animal trial to evaluate the efficacy of MSC treatment for PD comprised injecting ADSCs into the tunica albuginea. This led to a considerable improvement in erectile function, as well as an inhibition of the expression of type III collagen and elastin [37]. ADSCs with and without human IFN-b2b expression were injected into the tunica albuginea of a rat model for PD in a later investigation that was conducted by Gokce and colleagues. Regardless of whether or not IFN-b2b was present, the findings indicated a substantial improvement in erectile function as well as an attenuation of PD-like alterations [38]. In spite of these encouraging results in animal models, there is currently insufficient information regarding the efficacy of stem cell treatment for PD in people, which prevents its endorsement for clinical usage.

### 8. Discussion

Peyronie's disease is a debilitating condition characterized by the formation of fibrous plaques in the tunica albuginea, which can cause pain, curvature, and erectile dysfunction. While surgical intervention has traditionally been the curative option, the use of intraplaque injections of different substances has always been tested as a viable treatment option. One such substance is verapamil, a calcium channel blocker that has been shown to have antifibrotic and anti-inflammatory effects. Several studies have investigated the use of intralesional verapamil injections in patients with PD and have found it to be effective in reducing plaque size and improving penile curvature and erectile function.

Another substance that has been investigated for intralesional injections is interferon alpha-2b, a cytokine that has been shown to have antifibrotic and immunomodulatory effects. Studies have suggested that intralesional interferon alpha-2b injections may

lead to improvements in plaque size and penile curvature, as well as in penile pain and erectile function.

Collagenase, an enzyme that breaks down collagen, has also been used in intralesional injections for PD. The idea behind this treatment is to break down the fibrous plaque that is causing the curvature and other symptoms. Several studies have shown that collagenase injections can lead to improvements in plaque size and penile curvature, as well as in penile pain and erectile function.

The use of HA as a noninvasive alternative has gained attention in the last 30 years.

Preclinical studies have demonstrated the potential benefits of HA in treating PD. Such results are promising, but there is still much to learn about its mechanisms of action.

The antifibrotic, anti-inflammatory, and proangiogenic effects have been partially demonstrated; however, future studies should investigate more the molecular pathways involved in the inhibitory effects of HA on fibrosis.

Regarding the anti-inflammatory effects, it is interesting to note that HA has both structural and signaling roles in the body, and these roles can be influenced by various factors such as the molecular weight, location, and cell factors. The ability of HA to modulate tissue hydration and maintain the physical properties of ECM is important for the maintenance of healthy tissues. On the other hand, the signaling role of HA in inflammation and cell behavior highlights its potential in therapeutic applications in various conditions. However, more research is needed to elucidate the signaling pathways involved in HA's anti-inflammatory effects, as well as to optimize the delivery methods and the type of molecule that should be used for the treatment of PD.

Not many clinical trials have been conducted on the use of HA in patients suffering from PD; however, all of them have shown promising results. The studies included patients who received intralesional injections of HA over varying periods of time and measured outcomes such as plaque size, penile curvature, and erectile function. Overall, the studies found that HA injections led to improvements in these outcomes and were well tolerated by patients. Additionally, two studies compared the effectiveness of HA injections with another treatment, verapamil, and found that HA injections were more effective in reducing penile curvature and improving patient satisfaction. One study also investigated the combined use of oral and intralesional HA and found that this approach was more effective than intralesional HA injections alone.

Clearly, there are some limitations in the whole literature. In particular, the sample sizes of the studies. Several studies included a relatively small number of participants, which may limit the generalizability of the results to the broader population of individuals with PD. Moreover, in some studies, there is a lack of control groups which makes it difficult to determine whether the observed improvements were a result of the treatment or other factors such as natural disease progression or placebo effects. In addition, there is a huge variability in treatment protocols. In fact, there was a wide range of treatment protocols across the studies, such as different dosages (when elicited) and frequencies of injections, which may make it difficult to compare results across studies or determine the optimal treatment approach.

The findings suggest that HA injections could be a viable and safe treatment option for PD, particularly in the early stages of the disease. Regarding the stable phase of PD, the absence of studies cannot lead to any conclusions and, obviously, further research is needed to confirm these results and to determine the optimal dosage and treatment duration for HA injections, either in active or in chronic PD.

## 9. Conclusions

In conclusion, hyaluronic acid has shown promising results as a nonsurgical treatment option for Peyronie's disease, both in preclinical and clinical studies. Its ability to inhibit fibrosis and promote angiogenesis makes it a potentially effective therapy. Moreover, the low risk of adverse effects makes it an attractive option for patients who are unable or unwilling to undergo surgery. While more research is needed to fully understand its

efficacy and optimal dosage, the current evidence suggests that hyaluronic acid is a safe and effective treatment for Peyronie's disease.

**Author Contributions:** Conceptualization, M.C., R.L.R. and G.C.; methodology, M.C., R.L.R. and G.C.; software, M.C., R.L.R. and G.C.; validation, M.C., R.L.R. and G.C.; formal analysis, M.C., R.L.R. and G.C.; investigation, M.C., R.L.R. and G.C.; resources, M.C., R.L.R. and G.C.; data curation, M.C., R.L.R. and G.C.; writing—original draft preparation, M.C., R.L.R. and G.C.; writing—review and editing, M.C., R.L.R. and G.C.; visualization, M.C., R.L.R. and G.C.; supervision, M.C., R.L.R. and G.C.; project administration, M.C., R.L.R. and G.C.; funding acquisition, M.C., R.L.R. and G.C. All authors have read and agreed to the published version of the manuscript.

**Funding:** This research received no external funding.

**Institutional Review Board Statement:** Not applicable.

**Informed Consent Statement:** Not applicable.

**Data Availability Statement:** Not applicable.

**Conflicts of Interest:** The authors declare no conflict of interest.

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
