# Peer review of "Current Evidence on the Use of Hyaluronic Acid as Nonsurgical Option for the Treatment of Peyronie’s Disease: A Contemporary Review"

_2673-4397, doi:10.3390/uro3020017_

Round 1

Reviewer 1 Report

Minor comments:

1.    In the “Abstract” section, the authors are invited to define HA at first use or to avoid the use of contracted form.

2.     In the “Introduction” section, in “and pain during intercourse” I suggest to eliminate “during intercourse” since the pain could be referred also in the flaccid state.

3.     In the “Introduction” section, I suggest to cite the symptomatic burden of patients affected by PD as shown in PMID: 36426559.

4.     In the “Introduction” section, I suggest to the authors to declare the paragraph of the aim of the study, the will of providing also brief summaries on alternative non-surgical therapies for PD.

5.     In the “Introduction” section, “It is injected directly into the plaque and has been shown to improve penile curvature and reduce symptoms in clinical trials” needs a citation.

6.     In the whole manuscript, after defining an acronymous, the authors are invited to always use the contracted form; i.e., in the “Introduction” section with “Peyronie’s disease” in full form.

Author Response

Dear Reviewer 1,

Thank you very much for taking the time to review our paper. We know it takes a lot of time to review papers and we thank you for that. I will provide a point by point response to your review in order to have a clear communication.

  1. In the “Abstract” section, the authors are invited to define HA at first use or to avoid the use of contracted form.

We have changed that in the abstract. Thank you

2) In the “Introduction” section, in “and pain during intercourse” I suggest to eliminate “during intercourse” since the pain could be referred also in the flaccid state.

We have changed that in the introduction. Thank you

3)     In the “Introduction” section, I suggest to cite the symptomatic burden of patients affected by PD as shown in PMID: 36426559.

Absolutely correct. We have changed that in the References. Thank you

4)     In the “Introduction” section, I suggest to the authors to declare the paragraph of the aim of the study, the will of providing also brief summaries on alternative non-surgical therapies for PD.

Absolutely correct. We have added that at the end of the introduction. Thank you

5)     In the “Introduction” section, “It is injected directly into the plaque and has been shown to improve penile curvature and reduce symptoms in clinical trials” needs a citation.

We have added two references to that sentence. Thank you very much.

6)     In the whole manuscript, after defining an acronymous, the authors are invited to always use the contracted form; i.e., in the “Introduction” section with “Peyronie’s disease” in full form.

Absolutely right. We have checked the whole manuscript again. Thank you very much.

Reviewer 2 Report

Dear Author, in recent years articles have been published on the use of Pentoxifylline, Carnitine, Propolis etc. in Peyronie’s treatment, and other articles have reported the total regression of the disease after treatment with antioxidants. These articles are present in the literature and they still need to be cited and synthetically summarized in the Introduction section, to give complete information on the "state of the art" of Peyronie’s therapy.
I hope you understood the importance of this enrichment of your article. 

Author Response

Dear Reviewer 2

Thank you very much for taking the time to review our paper. We know it takes a lot of time to review papers and we thank you for that. I will respond to your only point to the review. 

1) “Dear Author, in recent years articles have been published on the use of Pentoxifylline, Carnitine, Propolis etc. in Peyronie’s treatment, and other articles have reported the total regression of the disease after treatment with antioxidants. These articles are present in the literature and they still need to be cited and synthetically summarized in the Introduction section, to give complete information on the "state of the art" of Peyronie’s therapy.

I hope you understood the importance of this enrichment of your article.“

We know there are a lot of papers reporting the treatment with antioxidants, Pentoxifylline, Carnitine, etc. However I did not cite them as according to the EAU/AUA Guidelines those are not recommended. Thus we decided voluntarily not to add them as we do agree with guidelines that such antioxidants should not be offered to patients with Peyronie’s Disease. 

Here it is the EAU citation : “Do not offer oral treatment with vitamin E, potassium para-aminobenzoate (potaba), tamoxifen, pentoxifylline, colchicine and acetyl esters of carnitine to treat Peyronie’s disease (PD).” Strength of rating : STRONG.

We hope you may understand our will.

Reviewer 3 Report

The manuscript by Cpece et.al. summarized recent literature on the use of hyaluronic acid in Peyronie's disease. Some comments:

1. The title of the manuscript is on the use of hyaluronic acid in Peyronie's disease however part 4 and 5 of the manuscripts described additional intra-lesional treatments that are irrelevant to the title. The authors should consider either removing the two parts or changing the title to make it more accurate to reflect contents of the manuscript

2. The authors included five clinical trials. The last two trials were done in patients in the acute phase of disease. Please clarify if the first three trials were also done in the acute phase of disease or not.

3. There are three subtitles numbered as "3" and two subtitles numbered as "5". Needs clarify. 

4. Page 4 line 9, "... in compared to..." should be  "in comparison to".

Author Response

Dear Reviewer,

thank you very much for your revision. I will respond to you point by point.

1) The title of the manuscript is on the use of hyaluronic acid in Peyronie's disease however part 4 and 5 of the manuscripts described additional intra-lesional treatments that are irrelevant to the title. The authors should consider either removing the two parts or changing the title to make it more accurate to reflect contents of the manuscript.

"Thank you very much. We did not focus on this very important point. We have indeed adjusted the title since this paper had the aim of focusing on the use of hyaluronic acid in the context of non surgical treatments."

2) The authors included five clinical trials. The last two trials were done in patients in the acute phase of disease. Please clarify if the first three trials were also done in the acute phase of disease or not.

"All studies have been performed in patients with acute phase of peyronie's disease. I have added this in all sentences."

3) There are three subtitles numbered as "3" and two subtitles numbered as "5". Needs clarify. 

"Absolutely correct. We have corrected that.

4) Page 4 line 9, "... in compared to..." should be  "in comparison to".

"Thank you very much. We have corrected that."

We hope the paper is complete thanks to your advice.

Kind Regards

Round 2

Reviewer 2 Report

The articles concerning the therapies with antioxidants are important beyond the guidelines and must be cited (in the Introduction section) and commented in any case. The article is not complete and the reviewer's recommendations have been rejected for reasons that are not completely scientific.

Author Response

Dear Reviewer,

thank you very much for your revision. I have added in the introduction the following sentence :

"Non-surgical treatments for Peyronie's disease include a variety of options, such as oral antioxidants therapies or intralesional injections. Antioxidants, such as vitamin E and coenzyme Q10, have been suggested to potentially reduce oxidative stress and inflammation, which are believed to play a role in the progression of Peyronie's disease. However, there is limited research specifically addressing the use of antioxidants in Peyronie's disease"

I hope you consider it complete.

Kind Regards